# Think Deep, Think Fast: Investigating Efficiency of Trained-verifier-free Inference-time-scaling Methods

## Abstract

There is intense interest in investigating how inference time compute (ITC) (e.g. repeated sampling, refinements, etc) can improve large language model (LLM) capabilities. At the same time, recent breakthroughs in reasoning models, such as Deepseek-R1, unlock the opportunity for reinforcement learning to improve LLM reasoning skills. An in-depth understanding of how ITC interacts with reasoning across different models could provide important guidance on how to further advance the LLM frontier. This work conducts a comprehensive analysis of inference-time scaling methods for both reasoning and non-reasoning models on challenging reasoning tasks. Specifically, we focus our research on trained-verifier-free inference time-scaling methods due to its generalizability without needing a reward model. We construct the trade-off curve of quality and efficiency. We find that non-reasoning models, even with an extremely high inference budget, still fall substantially behind reasoning models. For reasoning models, majority voting proves to be a robust inference strategy, generally competitive or outperforming other more sophisticated ITC methods like best-of-N and sequential revisions, while the additional inference compute offers minimal improvements. We further perform in-depth analyses of the association of key response features (length and linguistic markers) with response quality, with which we can improve the existing ITC methods. We find that correct responses from reasoning models are typically shorter and have fewer hedging and thinking markers (but more discourse markers) than the incorrect responses.

## 1 Introduction

Language models have witnessed remarkable advancements in recent years, demonstrating increasingly sophisticated capabilities across various tasks (OpenAI, 2023; Dubey et al., 2024; Bai et al., 2023). Despite these improvements, complex reasoning remains challenging, often requiring additional computational resources and specialized techniques to achieve satisfactory performance (Wang et al., 2024b; Yao et al., 2023). This challenge has motivated the development of inference-time compute (ITC) scaling methods, which allocate additional computational resources during inference to enhance model outputs.

The landscape of language model reasoning has evolved along two primary dimensions. First, approaches like Chain-of-thought (Wei et al., 2023), self-consistency (Wang et al., 2022), tree-structured sampling (Snell et al., 2024), and mixture of agents (Wang et al., 2025) have emerged as effective techniques for boosting reasoning capabilities during inference without requiring model parameter changes. Second, a new class of "reasoning models", explicitly post-trained to solve highly challenging problems, has been introduced, exemplified by models like o1 (OpenAI et al., 2024), Deepseek-R1 (DeepSeek-AI et al., 2025), and QwQ (Team, 2024).

While both approaches show promise, they introduce significant computational overhead. Chain-of-thought Wei et al. (2023) prompting increases token generation, tree-structured sampling requires exploring multiple solution paths (Liu et al., 2025), and mixture of agents (Wang et al., 2025) demands running several specialized agent configurations simultaneously. This computational burden raises critical questions about efficiency: how can we optimize the trade-off between computational

resources and reasoning performance? Which inference-time scaling methods deliver the best results for different model architectures? How do reasoning models compare to conventional models under varying computational budgets?

These questions remain largely unsolved in the current literature. Prior work has primarily focused on fine-tuning strategies (Yu et al., 2025; Xie et al., 2025) or simple inference evaluations (Brown et al., 2024a), with limited attention to systematic evaluation of inference-time compute scaling across model types. Furthermore, the emergence of specialized reasoning models demands a thorough reassessment of established inference-time methods, as these models may respond differently to such techniques compared to their non-reasoning counterparts. Our work tries to bridge this gap by conducting a comprehensive analysis of trained-verifier-free inference-time scaling methods across both reasoning and non-reasoning models. We specifically focus on trained-verifier-free approaches due to their broader applicability—they do not require specialized reward models or additional training, making them more accessible and generalizable across different domains and applications.

Through extensive experimentation, we construct trade-off curve that reveal efficiency-performance trade-offs, demonstrating that majority voting—particularly when weighted by reasoning length—provides an optimal balance between computational cost and reasoning performance. Furthermore, we demonstrate that conventional non-reasoning models, even with substantial inference-time budgets, consistently underperform compared to reasoning models. For reasoning models, our analysis reveals that sophisticated methods like mixture-of-agents, best-of-N, and sequential revisions offer minimal improvements over the simple majority-voting baseline. Our research provide the following **key contributions**:

- We provide a comprehensive comparison of inference-time scaling methods across reasoning and non-reasoning models, establishing trade-off curve for efficiency-performance trade-offs.

- We demonstrate that the simple majority-voting method consistently outperforms other more complex ones, challenging the notion that advanced inference-time techniques are necessary to improve model's reasoning behaviors.

- We reveal that non-reasoning models remain fundamentally limited in reasoning tasks regardless of inference-time compute budget, highlighting the intrinsic value of models specifically designed for reasoning.

- We analyze the correlation between response linguistic features (response length, linguistic markers) and task performances, providing practical guidance for improving existing inference-time methods without increasing computational costs.

## 2 RELATED WORK

### 2.1 INFERENCE-TIME SCALING

Recent work has demonstrated that scaling compute during inference offers a promising alternative to costly model pretraining. Language models (Touvron et al., 2023; Jiang et al., 2023a; Team et al., 2024) continue to improve with increased data and parameters, though at escalating development costs. Inference-time scaling approaches like Brown et al. (2024b) show a log-linear relationship between problem-solving coverage and sample count across reasoning tasks, offering cost-effective alternatives to model size scaling. Inference-time architectures combine techniques such as generation ensembling, sampling, ranking, and fusion to exceed individual model performance. Works including Mixture-of-Agents (Wang et al., 2024c), LLM Blender (Jiang et al., 2023b), and orchestration frameworks like DSPy (Khattab et al., 2023) demonstrate these approaches' effectiveness. Even with single models, techniques like chain-of-thought (Wei et al., 2023) and branch-solve-merge (Saha et al., 2024) enhance reasoning capabilities. Our work extends this literature by focusing specifically on trained-verifier-free inference-time scaling methods that don't require additional reward models. We systematically evaluate these methods across both reasoning-specialized models (e.g., DeepSeek-R1) and general-purpose LLMs (e.g., Llama, Qwen) on complex reasoning benchmarks including MATH, AIME, GPQA, and LiveCodeBench. Our findings align with and extend those of Wang et al. (2024c) by demonstrating that majority voting—particularly when weighted by

reasoning length—provides optimal balance between computational cost and performance. Unlike prior work that primarily focused on technique development, we construct comprehensive trade-off curve revealing several key insights: (1) non-reasoning models, even with substantial inference-time budgets, consistently underperform reasoning-specialized models; (2) sophisticated methods like sequential revisions offer minimal improvements for reasoning models over simpler majority voting; and (3) response features like length and linguistic markers provide valuable signals for identifying high-quality responses. These findings provide practical guidance for balancing reasoning quality and computational efficiency across model architectures and deployment scenarios.

## 2.2 INFERENCE EFFICIENCY

Recent work has explored inference-time optimization strategies that trade additional computation for improved model performance. A key approach is repeated sampling and majority voting, leveraging LLMs' generative diversity (Brown et al., 2024b; Wang & Li, 2024). To reduce computational cost, refinements like Confidence-Informed Self-Consistency (CISC) use confidence-weighted voting, cutting required samples by over 40% (Zhang & Kumar, 2024). Another strategy, DivSampling, injects prompt perturbations to increase answer diversity, boosting performance across math, reasoning, and code generation (Liu & Chen, 2024). These methods illustrate a broader trend: sacrificing more inference compute (e.g., multiple forward passes) for superior results.

Another active direction integrates Monte Carlo tree search (MCTS) with LLMs to enhance inference-time exploration. MCT-Self-Refine (MCTSr) combines iterative tree search and self-evaluation, significantly improving math problem-solving (Lee & Kim, 2024). Adaptive Branching MCTS (AB-MCTS) refines search by dynamically deciding to explore new candidates or refine existing ones, outperforming fixed-branch MCTS on coding and engineering tasks (Garcia & Martinez, 2024). Beyond traditional NLP, MCTS has been applied to agentic settings: a text-based game agent equipped with memory-augmented MCTS planning achieved higher scores by learning from past trials (Patel & Sharma, 2024).

## 3 METHODOLOGY

### 3.1 MODELS

The study evaluates a diverse set of models to cover a wide range of model sizes and architectures, crucial for understanding the effectiveness of ITC methods across different capabilities. The models are categorized into non-reasoning and reasoning models, reflecting their primary strengths and training focus.

**Non-reasoning models** Non-reasoning models are general-purpose LLMs optimized for tasks like text generation and dialogue, but they may lack specialized training for complex reasoning. The selected models include: GPT-4o-mini, Qwen2.5-7B-Instruct, Qwen2.5-72B-Instruct (Bai et al., 2023), Llama-3.3-70B-Instruct, and Llama-3.1-8B-Instruct (Dubey et al., 2024). This selection covers a wide range of sizes (from 8B to 72B parameters) and includes both open-source and closed-source models, ensuring a comprehensive evaluation. The inclusion of GPT-4o-mini as a closed-source model contrasts with the open-source Qwen2.5 and Llama series, highlighting the study's intent to assess performance across different accessibility models.

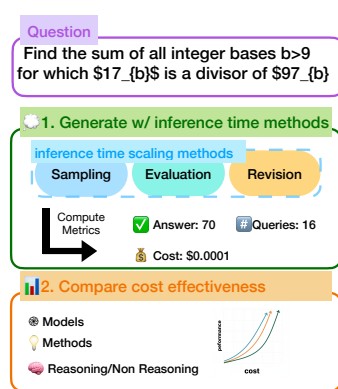

Figure 1: We analyze the efficacy of various reasoning methods and models.

**Reasoning models** Reasoning models are specifically trained or designed to handle complex reasoning tasks, such as mathematical problem-solving and code generation, often through methods like reinforcement learning (RL). The selected models include: DeepSeek-R1-Distill-Llama-

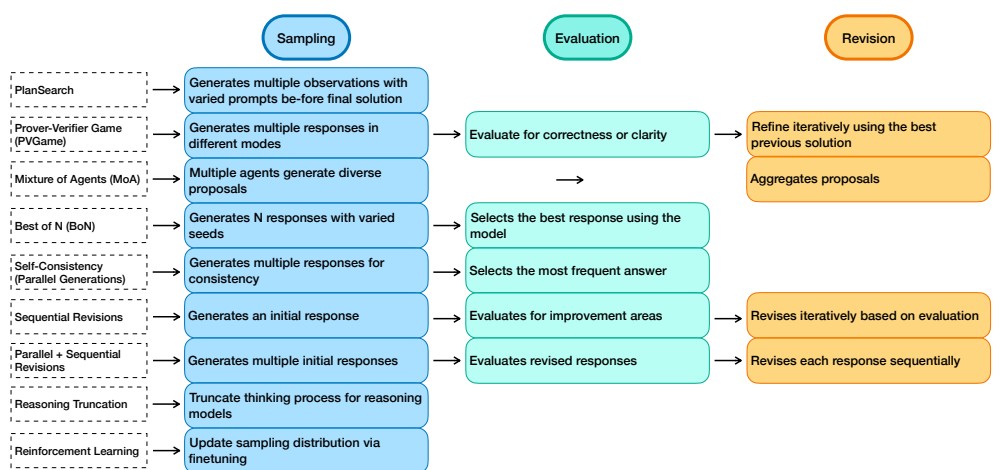

Figure 2: Inference-Time Scaling Methods: Sampling, Evaluation, and Revision approaches. Note that we view reinforcement learning and reasoning truncation as an approach to change the sampling distribution.

70B (DeepSeek-AI et al., 2025), DeepSeek-R1-Distill-Llama-8B, DeepSeek-R1-Distill-Qwen-32B, DeepSeek-R1-Distill-Qwen-14B, DeepSeek-R1-Distill-Qwen-7B, and QwQ-32B-Preview (Team, 2024). For some experiments we also evaluate DeepSeek-R1 (DeepSeek-AI et al., 2025), but not all due to the high-cost nature of inference-scaling methods.

## 3.2 TASKS

**MATH 500**   MATH 500 is a subset of challenging competition mathematics problems from the MATH dataset (Hendrycks et al., 2021). The dataset contains complex high school math problems that are often solved with step-by-step reasoning.

**AIME**   AIME consists of problems from the American International Mathematics Examination (AIME), a prestigious mathematics competition for high school students. AIME is known for its difficult and thought-provoking problems and we select the 2024 subset as our evaluation benchmark which contains 30 problems.

**GPQA**   Graduate-Level Google-Proof Q&A Benchmark (GPQA) (Rein et al., 2023), is a dataset of 448 multiple-choice questions in biology, physics, and chemistry, crafted by domain experts. It is designed to be extremely challenging, with experts who have or are pursuing PhDs in the corresponding domains reaching only 65% accuracy. We use GPQA Dimaond which is a high quality subset.

**LiveCodeBench**   Livecodebench (Jain et al., 2024) provides a holistic and contamination-free assessment of large language models for code-related tasks. The code generation (codegen) subtask, specifically, is selected in this work to test the models' ability to generate correct code for these problems. We use the code problems from 2024-11-01 to 2025-02-01.

## 3.3 INFERENCE-TIME SCALING METHODS

Inference-time scaling methods in language models leverage additional computational resources during the inference phase to enhance performance by adaptively modifying the model's output distribution for a given prompt at test time. This process involves altering how responses are generated and processed to achieve more accurate or complex outputs compared to direct sampling from the model. The study categorizes these methods into three key steps: sampling, evaluation, and revision, each defined as follows:

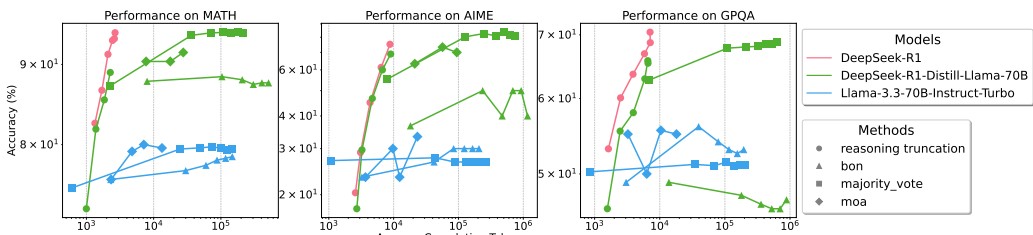

Figure 3: The overview of inference-time-compute methods for reasoning and non-reasoning models. Even though inference-time scaling method improves Llama-3.3-70B, it still struggles to beat the R1-distilled version of Llama 70B. However, with very limited compute, non-reasoning model with inference method can be at the trade-off curve.

**Sampling** is the process of generating one or more responses from the language model for a given prompt, potentially with modifications to the input to influence the distribution from which samples are drawn. In inference-time scaling, sampling goes beyond simply drawing responses from the model's default distribution. Modifications at the input level—such as adding specific tokens or phrases (e.g., in chain-of-thought prompting)—shift the distribution to favor responses that align with desired reasoning outcomes.

**Evaluation** is the process of assessing the quality or correctness of the generated responses, which informs how we modify the output distribution by selecting, weighting, or ranking the responses. Once responses are sampled, evaluation determines their value, often using metrics like accuracy, coherence, or task-specific scores given by human, reward models or LLMs. In the context of inference-time scaling, this step shapes the output distribution by concentrating it on higher-quality responses. In this work all verifiers used are LLMs, meaning the same model is used for both sampling and evaluation, which can introduce biases but enhances generalizability without requiring external rewards.

**Revision** is the process of modifying or improving the generated responses based on the evaluation, potentially involving iterative refinement or generation of new responses, thereby transforming the output distribution to concentrate on higher-quality outputs.

We selected several representative methods that enhance language model performance by adaptively modifying the model's output distribution at test time, using additional compute during inference (Figure 2). Each method leverages sampling (generating responses), evaluation (assessing response quality), and revision (refining responses) in distinct ways to achieve this goal. Specifically, we picked PlanSearch (Wang et al., 2024a), Prover-Verifier Game (PVGame) (Kirchner et al., 2024), Mixture of Agents (MoA) (Wang et al., 2025), Best of N (BoN), Self-consistency (Wang et al., 2022), Sequential Revisions, and Parallel + Sequential Revisions (Snell et al., 2024). More details can be found in the Appendix.

## 4 EXPERIMENTS AND ANALYSIS

### 4.1 CONSTRUCTING THE TRADE-OFF CURVE OF QUALITY AND EFFICIENCY

In this section, we will give an overview of various inference-time scaling methods and present the current trade-off curve for those methods. We aim to provide a comprehensive view and provide guidelines for the research question: what model or method should we use with a given inference budget?

#### 4.1.1 NON-REASONING MODELS AUGMENTED BY INFERENCE-TIME METHODS CANNOT MATCH REASONING MODELS

Our comprehensive analysis reveals critical insights into the performance of reasoning and non-reasoning models under various inference-time scaling methods. Figure 3 demonstrates a compelling finding: non-reasoning models, even when enhanced by advanced inference-time scaling

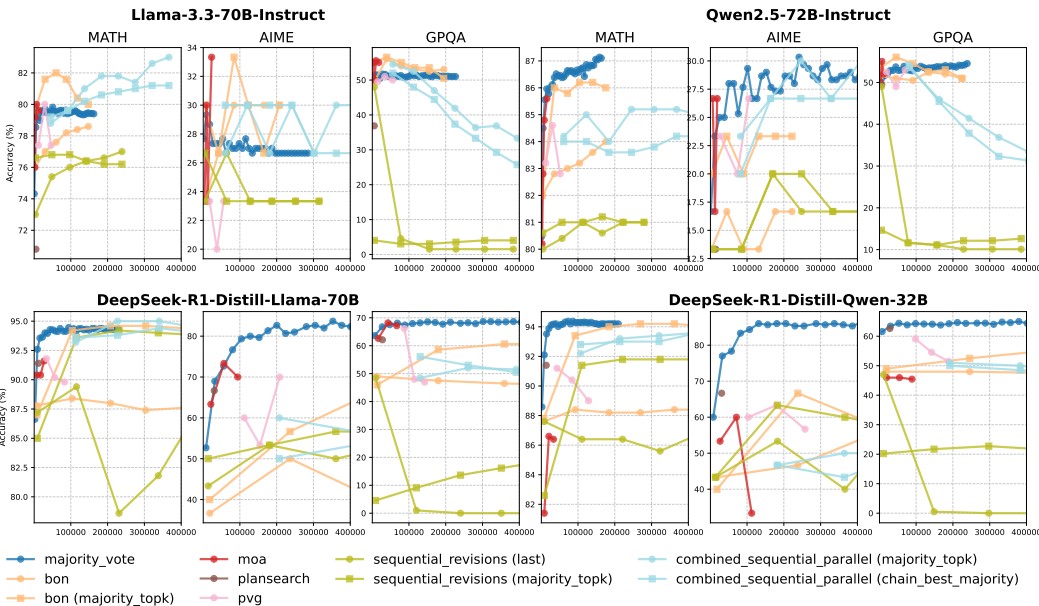

Figure 4: Performance of various inference-time scaling methods for four models across MATH, AIME and GPQA. For some methods we have multiple eval metrics. For *bon* (best-of-N approach), we pick the highest scored response. *majority topk* means we use top k scored resposnes to do majority voting (we always set k as half of total samples). *last* means we pick the last revisioned sample. *chain best majority* indicates that we use the best scored sample from each chain and then take majority. Due to high cost of inference, methods like sequential revisions and combined sequential parallel are only sampled once, which may seem volatile when plotted. The results for other models can be found in the Appendix.

techniques, consistently underperform compared to reasoning models. Specifically, we observed that R1-Distilled versions of Llama-3.3-70B significantly outperform their original Instruct counterparts. Despite employing sophisticated inference-time scaling methods, non-reasoning models fail to match the performance of purpose-built reasoning models. This empirical evidence suggests that for compute-optimal approaches, investing in training specialized reasoning models may provide substantially better long-term efficiency compared to repeated inference-time scaling of general models. Note that in Figure 3, we added another baseline "reasoning truncation" by simply truncating the reasoning process (encapsulated in "⟨think⟩" "⟨/think⟩" tokens) and then extended the response to the end by sequence completions. We found that this can significantly degrade the response quality, and extreme truncation eventually crossed the curve of non-reasoning models.

### 4.1.2 TRAINING-FREE TRAINED-VERIFIER-FREE METHODS DOES NOT CONSISTENTLY HELP REASONING MODELS

Training-free, verifier-free inference-time scaling methods offer minimal improvements for reasoning models. As shown in Figure 4, these methods—which do not require specialized reward models or additional training—fail to significantly enhance the performance of reasoning models. Almost all the methods are underperforming majority voting for both DeepSeek-R1-Distill-Llama-70B and DeepSeek-R1-Distill-Qwen-32B. This observation suggests that reasoning models, by design, already incorporate sophisticated reasoning capabilities during their training phase. This conclusion is less clear for non-reasoning models. Both sequential revisions and combined sequential parallel method outperform majority vote onall tasks for Llama-3.3-70B-Instruct. However, for Qwen2.5-72B-Instruct, we only still see that majority voting still outperforms most methods.

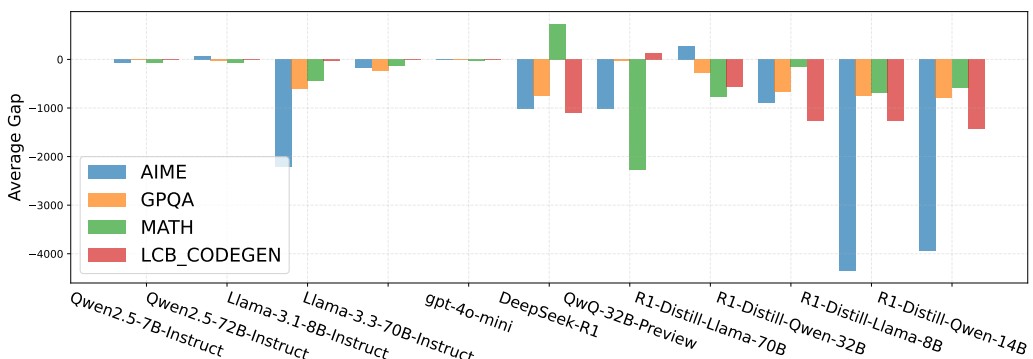

Figure 5: The average response length gap for each model tasks across four tasks. The average response length gap is computed by: 1) calculating mean length difference between correct and incorrect responses within each question and 2) averaging these differences across the entire dataset. LCB_CODEGEN represents the code generation subtask in the LiveCodeBench benchmark.

### 4.1.3 MAJORITY VOTING IS OFTEN THE BEST INFERENCE-TIME SCALING METHOD – FOR BOTH REASONING AND NON-REASONING

Across both reasoning and non-reasoning models, majority voting consistently demonstrates superior performance. The method's simplicity belies its effectiveness, as illustrated in Figure 4, where it outperforms more complex inference-time scaling approaches such as mixture-of-agents, best-of-N, sequential revisions and combined parallel sequential. The success of majority voting can be attributed to its fundamental approach of leveraging multiple model outputs. By aggregating responses, the method effectively mitigates individual model biases and captures a more robust representation of the underlying reasoning.

### 4.2 THE IMPACT OF RESPONSE LENGTH OF MODEL AND TASK PERFORMANCE

The relationship between response length and correctness in reasoning models remains contentious in recent literature. Some studies (Chen et al., 2024) suggest that reasoning models generate unnecessarily verbose responses, even for simple problems. This section aims to systematically investigate the correlation between response length and model performance, providing empirical insights into how reasoning models generate and structure their responses. Our analysis seeks to demystify existing observations by rigorously examining response characteristics across various reasoning tasks and model architectures. By leveraging comprehensive datasets, we will explore the nuanced interplay between response length, model design, and task complexity. Note that in order to remove query difficulty as a confounder, we compare length across different samples from the *same* query. That is, we only compare samples length within the same query not across different queries.

### 4.2.1 NON-REASONING MODELS SHOW NO TREND

Non-reasoning models present an absence of clear correlation between response length and correctness. Figure 5 plots the average response length gap for each model tasks across four tasks. The average response length gap is computed via first the mean length difference between correct and incorrect responses within individual questions (There are 100 samples for resoning models and 256 samples for non-reasoning models). Then we average these differences across the entire dataset. We discard the questions that only include one class of correctness (all correct, or all incorrect). This ensures that the average gap is not biased by questions that are too easy or too difficult. It reveals that the response length gaps across all four datasets are low for all non-reasoning models, with an exception of Llama-3.1-8B-Instruct. In that case, we observe a non-negligible gap for the AIME task.

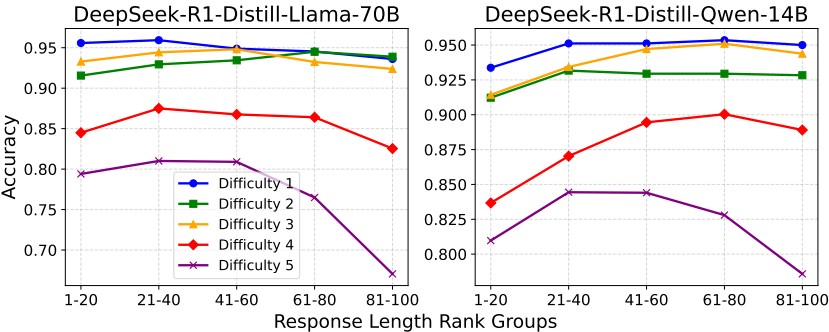

Figure 6: Accuracy of responses in different length groups. For each question, we generate 100 samples and then we bin those samples into five bins. Then average accuracy is computed for each bin across the dataset.

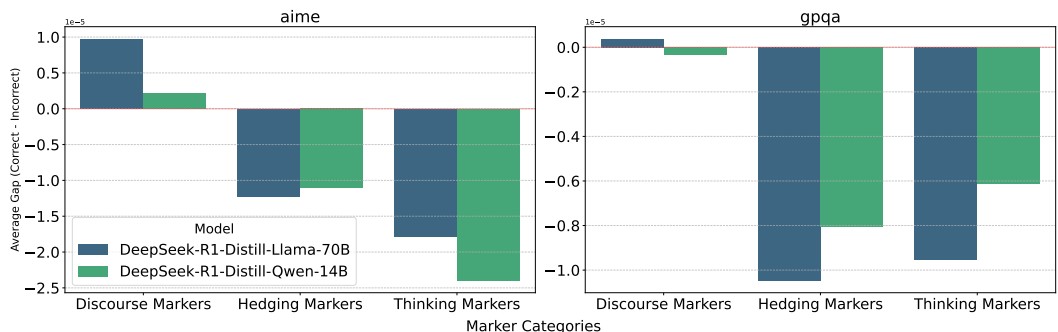

Figure 7: Average gaps between correct and incorrect responses for DeepSeek-R1-Distill-Llama-70B and DeepSeek-R1-Distill-Llama-14B. The average gaps are first computed by using computing the mean difference of thinking token frequency of correct and incorrect responses within one question and then average over the entire dataset. The frequency is weighted by response length. Refer to Table 1 for the definition of different marker categories.

### 4.2.2 FOR REASONING MODELS, SHORTER RESPONSES ARE MORE LIKELY TO BE CORRECT

Unlike non-reasoning models, reasoning models show a clearer trend of shorter, more precise responses being more accurate. Figure 5 provides clear evidence of this inverse relationship between response length and accuracy. This phenomenon reflects the sophisticated reasoning mechanisms inherent in these models. Different reasoning models behave differently for each task. However, there is a consistently largeer gap for the AIME task. We will go into more details in next section.

### 4.2.3 CORRECTNESS GAP WIDENS WITH INCREASING PROBLEM DIFFICULTY

Figure 5 reveals a critical insight into model performance across varying task complexities. As problem difficulty increases, the gap between correct and incorrect responses becomes more pronounced, particularly for reasoning models. The AIME dataset, known for its challenging nature, exemplifies this trend, with all reasoning models demonstrating a wider correctness gap. To systematically investigate this phenomenon, we analyze response length differences using the MATH dataset, which offers a natural difficulty gradient ranging from level one to five. We stratify samples for each question into five bins ranked from shortest to longest responses. We find that for high-difficulty problems (level 5), where an inverse relationship between response length and correctness became evident. Specifically, Figure 6 demonstrates that as problem complexity increases, reasoning models tend to generate more accurate responses with shorter lengths.

### 4.3 LINGUISTIC MARKERS AND WORD FREQUENCY ANALYSIS

Reasoning models tend to use certain linguistic markers, especially thinking tokens such as "alternatively" or "however". In this section, we investigate the relationships between such linguistic markers and correctness.

#### 4.3.1 LINGUISTIC MARKERS OCCURS MORE FREQUENTLY IN INCORRECT RESPONSES

A compelling finding emerges from our linguistic marker analysis: incorrect responses consistently exhibit a higher density and diversity of linguistic markers. Figure 7 provides empirical evidence demonstrating that both hedging and thinking markers (more details of marker definition can be found in Table 1 of Appendix) are markedly more prevalent in incorrect responses compared to correct ones. Our methodological approach involved computing the average marker frequency gaps by first calculating the mean difference in thinking token frequencies between correct and incorrect responses within individual questions, and then averaging these differences across the entire dataset. To account for response variability, we normalized the marker frequencies by response length. Note that this normalization will naturally result in very low frequency. Normally the marker counts range between 5 and 30 per response. The proliferation of markers in incorrect responses suggests a fundamental characteristic of ineffective reasoning. Rather than representing depth or complexity, the increased marker density appears to be a signal of cognitive imprecision—a tendency toward verbose and less focused responses. This observation provides a promising diagnostic tool for identifying potentially incorrect model outputs, suggesting that the quality of reasoning can be partially assessed through the strategic use and density of linguistic markers. The list of extra markers can be found at Table 1 in the Apprendix.

#### 4.3.2 LINGUISTIC MARKERS ARE A GOOD PREDICTOR OF RESPONSE CORRECTNESS

The notable discrepancy in marker counts between correct and incorrect responses motivated a further investigation into the predictive potential of linguistic markers. We designed a rigorous experimental framework to explore this hypothesis. Our method involved creating a dataset with extensive model response samples, generating 100 response generations per question, and then training a accuracy classifier based on the response features. We extracted marker features as input variables and response correctness as the target label. To ensure the robustness of model evaluation, we implemented a standard 0.6/0.2/0.2 train-validation-test split. The results show great promises. For the DeepSeek-R1-Distill-Llama-70B model, a simple logistic regression classifier achieved a test F1 score of 0.7469. The performance was even more impressive for the DeepSeek-R1-Distill-Llama-14B model, with an F1 score of 0.8637. Our finding suggests the potential of linguistic markers as a diagnostic tool for assessing model reasoning capabilities. The approach offers a nuanced method for evaluating model performance beyond traditional length metrics, opening new avenues for model interpretation and quality assessment.

## 5 CONCLUSION

Our work thoroughly assesses trained-verifier-free inference-time scaling methods for LLMs, emphasizing their efficiency and effectiveness in reasoning tasks. We found that non-reasoning models, despite leveraging advanced scaling techniques and significant computational resources, consistently lag behind specialized reasoning models like R1-Distilled Models. For reasoning models, simpler strategies such as majority voting often surpass more intricate methods like best-of-N or sequential revisions in performance. We also realize that correct responses are typically shorter and feature fewer linguistic markers (e.g., hedging terms), indicating these traits could serve as predictors of accuracy. Leveraging these response characteristics and linguistic marker features to enhance inference methods can be an intriguing future direction.

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

| Category | Examples |
|---|---|
| **Discourse Markers** | *on the other hand*, *nevertheless*, *moreover*, *in addition*, *furthermore*, *therefore*, *consequently*, *as a result* |
| **Hedging Markers** | *perhaps*, *maybe*, *possibly*, *it seems*, *might*, *could* |
| **Thinking Markers** | *however*, *wait*, *alternatively*, *hmm* |

Table 1: Categories of Linguistic Markers. Italicized words represent specific markers.

## A  USAGE OF LLM

We utilize LLM to assist some of the paper writing, code development and figure generation.

## B  INFERENCE-TIME SCALING METHODS

In this section, we will go into more detils about each inference-time scaling methods and how they are parametrized in our studies.

**Majority Vote**   Majority vote (self-consistency) generate multiple samples and choose the most frequent answer as final solution. Note that this method doesn't quite work for free-form generation problems such as LiveCodeBench. Hence we don't present results for LiveCodeBench. For reasoning model, we sample 100 samples for each query while for non-reasoning models, we sample 256 samples. For reasoning models, 100 samples were used due to higher computational cost and our observation that performance trends stabilized at this point. Non-reasoning models, being less computationally intensive, used 256 samples for greater robustness.

**PlanSearch**   This method prompts LLM to generate a number of observations and derived observations before making a final solution. We set number of generations to be three and number of derived observations to be two across all experiements.

**PVGame**   Prover-Verifier Games involves two main phases: solution generation and solution verification. Solutions are generated in both "helpful" and "sneaky" modes, evaluated for correctness and clarity, and refined iteratively by leveraging the best-scored solutions from prior rounds to guide subsequent attempts. In our setup, we fix the number of solutions each round to be three, and scale the number of rounds from one to three.

**Best of N**   Best of N samples N generations and each generation is evaluted via a judge. We use LLM as a judge and the judge template is shown in LLM evaluator prompt section below. We evaluate three times for each question and then take the mean as the final score.

**Sequential Revisions**   We follow the implementation of Snell et al. (2024). This method samples solution sequenially and each time it is revising from last solution except the first solution. The revision process involves first prompting feedback and then asking LLM to provide a revision based on those feedbacks. The feedback prompt template and the revision template first prompt is detailed in the LLM revision feedback prompt section and LLM revision prompt below respectively. Each solution is also evaluted using LLM as a judge. This is for choosing the final solution from the samples.

**Parallel + Sequential Revisions**   We again follow the implementation of Snell et al. (2024). This method samples multiple generations in the first step. For each geneneration, it sequentially revise similar to sequential revision independently. Same prompts are used from sequential revisions.

**Reasoning Truncation**   We truncate the reasoning process (encapsulated in "⟨ think⟩" "⟨/think⟩" tokens) to control for the budget.

## C  TEMPLATES

### C.1  LLM EVALUATOR PROMPT

---
**LLM evaluator prompt**

Question: **{question}**
Response: **{response}**

Analyze this answer strictly and critically. Identify and point out every flaw and imperfection to deduct the appropriate amount of points. Be very harsh and stringent in your assessment to ensure the grades are authoritative and reliable. Never award full marks. Assign a score between -100 and +100.

Response format:
[Analysis] ...
[Score] a single integer between -100 and +100.

---

### C.2  LLM REVISION FEEDBACK PROMPT

---
**LLM revsion feedback prompt**

Since we have a weak Answer, could you provide me with a relection or feedback to correct this answer better? Analyze this Answer Strictly and Critic, point out every flaw for ervery possible imperfect to minus every possible score! Let's think step by step.

Question: **{question}**

Answer to analyze: **{previous response}**

---

### C.3  LLM REVISION PROMPT

---
**LLM revsion prompt**

Please refine the your answer according to your Reflection or Feedback. The response should begin with [reasoning process]...[Verification]... and end with end with "[Final Answer] The answer is
boxed{answer}"
Let's think step by step.

Question: **{question}**

Previous solution: **{previous response}**

Feedback: **{feedback}**

---

## D  MORE INFERENCE TIME SCALING RESULTS

In this section, we present results for more models: Meta-Llama-3.1-8B-Instruct, Qwen2.5-7B-Instruct, gpt-4o-mini, DeepSeek-R1-Distill-Qwen-14B (Figure 6).

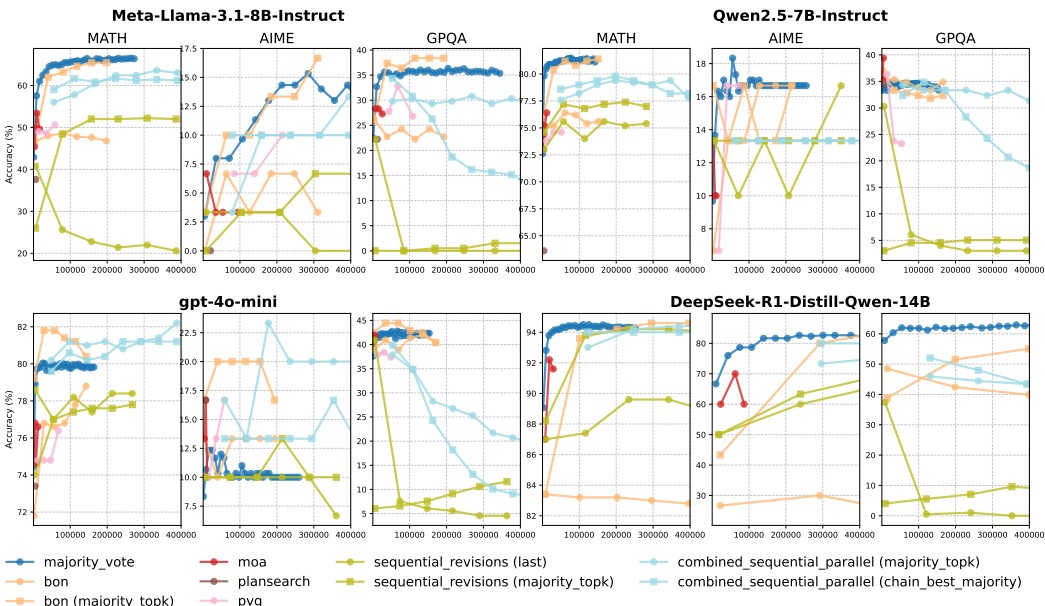

Figure 8: Performance of various inference-time scaling methods for four models across MATH, AIME and GPQA.

