# OpenReview forum: "Think Deep, Think Fast: Investigating Efficiency of Trained-verifier-free Inference-time-scaling Methods"
_ICLR.cc/2026/Conference — ICLR 2026 Conference Withdrawn Submission_

### Official Review · Reviewer_1wN4 · 2025-10-23

**Soundness:** 2
**Presentation:** 1
**Contribution:** 2
**Rating:** 2
**Confidence:** 4

**Summary:**

The paper tries to have a comprehensive evaluation for existing reasoning model and non-reasoning models in terms of inference-time scaling methods, response lengths and potential reasoning behaviors. It reviews some representative inference-time methods like majority voting and best-of-n, with several models such as  Llama-3.3-70B-instruct and Qwen-2.5-72B-instruct. It is important to have such comprehensive evaluation and draw some guidelines to inspire future research. This work found that simple majority-voting method is very promising and some simple correlation between response linguistic features and task performance.

**Strengths:**

1. The main experimental results are comprehensive and could reveal something.

**Weaknesses:**

1. it is hard-to-follow, including the textual description, figure presentation.
2. lots of analysis experiments are not comprehensive enough i.e., figure 6 and 7 only test some models, and it is not convincing to draw some conclusion rushly.
3. there is almost no consistent and interesting conclusions.

**Questions:**

1. figure 5: maybe just because lrm tends to use more lengthy response, which amplify the pattern.
2. 4.2.2 and 4.2.3 should be figure 6?
3. the conclusion at 4.2.2: training-free trained-verifier-free methods does not consistently help reasoning models also extend to non-reasoning models. there is no clear pattern for both types of models.
4. do you have direct answer for the research question in line 263: what model or method should we use with a given inference budget?
5. why figure 3 does not consider all methods and why some methods are missing from figure?

---

### Official Review · Reviewer_fA9U · 2025-10-30

**Soundness:** 2
**Presentation:** 3
**Contribution:** 1
**Rating:** 2
**Confidence:** 4

**Summary:**

This paper conducted a comprehensive empirical study to compare trained-verifier-free inference-time scaling (ITC) methods across both non-reasoning and reasoning LLMs. This work evaluated a range of ITC strategies on multiple reasoning benchmarks, and discovered several interesting findings, and the authors proposed to leverage these findings to improve ITC efficiency without additional compute.

**Strengths:**

1. This paper addresses explored a critical question in LLM research: how do inference-time scaling methods interact with model reasoning/non-reasoning nature? Empirical investigations and analysis on reasoning LLMs like OpenAI-o1 and DeepSeek-R1 are valuable for later research.

2. This work performed comprehensive analysis experiments with a diverse set of models (open/closed-source, varying sizes), multiple ITC methods, and several challenging reasoning benchmarks, making the findings more general and convincing.

**Weaknesses:**

1. This work compared and analyze “reasoning” and “non-reasoning” models but didn’t clearly define what features endowed LLMs “reasoning”. Is only the long Chain-of-Thought reasoning? What about the earlier short CoT (compared to o1- and R1-like), and even the hidden CoT? There is no clear definition for that.

2. All analysis in this paper does not include causal intervention experiments, for example, Section 4.2 didn’t not force to control the model to generate longer or shorter responses to observe resulting accuracy fluctuations. Similarly, it cannot establish whether the conciseness discussed in Section 4.3 is a cause rather than a consequence of correctness.

3. As the authors conducted extensive experiments and observed some interesting phenomena, this paper didn’t present any corresponding intervening approaches to take advantages of these findings, which undermined the contributions of this work severely.

4. The test-time scaling methods (as shown in Figure 2) can introduce unnecessary bias, for example, a model may consistently overrate verbose, marker-heavy responses it generates. The corresponding fixed prompts in Appendix C didn’t eliminate such a systemic bias.

5. The implementation details in this paper are not enough for reproduction, and there is no necessary codes or supplementary materials.

**Questions:**

1. Why is the majority voting method the most effective for reasoning models? And how could future researchers exploit such a finding to address the expensive reasoning costs?

2. All experiments employed fixed prompts. How sensitive are these findings to specific prompt format?

3. Simply truncation may delete the final answer, which may damage the accuracy of reasoning models, the authors should consider to substitute with better method to measure the budget trade-off.

---

### Official Review · Reviewer_eGrs · 2025-10-31

**Soundness:** 3
**Presentation:** 2
**Contribution:** 2
**Rating:** 2
**Confidence:** 3

**Summary:**

This paper presents a comprehensive empirical study of training-free, verifier-free inference-time scaling methods for large language models. The authors systematically evaluate strategies such as majority voting, best-of-N, and sequential revisions across reasoning and non-reasoning models on challenging benchmarks, including MATH, AIME, and GPQA. By constructing efficiency–performance trade-off curves, the study finds that (1) reasoning-specialized models (e.g., DeepSeek-R1 and its distilled variants) substantially outperform general-purpose models, even when the latter employ large inference-time compute budgets; (2) simple majority voting consistently matches or surpasses more complex inference strategies; and (3) correct responses from reasoning models tend to be shorter and contain fewer hedging or "thinking" linguistic markers than incorrect ones. These findings provide strong evidence that investing in specialized reasoning model training yields superior long-term efficiency compared to relying solely on inference-time computation, offering critical insights for future model development and deployment.

**Strengths:**

1. The paper conducts extensive experiments across multiple models (reasoning-specialized and general-purpose) and several challenging benchmarks (MATH, AIME, GPQA, LiveCodeBench), which provides a solid empirical basis for its claims.

2. The discovery that simple majority voting often matches or outperforms more complex inference-time strategies is actionable and immediately useful for practitioners optimizing inference budgets.

3. By evaluating multiple trained-verifier-free inference-time scaling methods within a single, consistent framework, the work reduces fragmentation in prior evaluations and enables fair cross-method comparisons.

**Weaknesses:**

1. The work builds on existing inference-time scaling methods rather than introducing a new algorithm. To strengthen impact, the authors should more clearly articulate the novel empirical insights—such as why majority voting excels in verifier-free settings—and how these shift prior understanding.

2. The evaluation focuses on mathematical and scientific reasoning. Extending it to commonsense or open-ended reasoning tasks would improve generalizability.

3. The link between linguistic cues and correctness is correlational. Controlled interventions (e.g., prompting to suppress hedging) could help establish causality and enhance interpretability.

**Questions:**

1. The correlation between linguistic cues and correctness is intriguing. Have the authors considered testing causal effects (e.g., by perturbing or masking such markers) to verify whether these features directly influence model reasoning accuracy?

2. How might the findings inform practical model deployment strategies—for example, deciding when to use majority voting versus more complex inference-time methods under constrained budgets.

---

### Official Review · Reviewer_xiiX · 2025-11-01

**Soundness:** 3
**Presentation:** 3
**Contribution:** 3
**Rating:** 6
**Confidence:** 4

**Summary:**

This paper comprehensively analyzes trained-verifier-free inference-time compute (ITC) scaling methods across LRMs (e.g., DeepSeek-R1, QwQ-32B-Preview) and LLMs (e.g., GPT-4o-mini, Llama-3.3-70B-Instruct) on tasks like MATH, AIME, GPQA, and LiveCodeBench. It constructs quality-efficiency trade-off curves, finding non-reasoning models, even with high ITC budgets, that underperform LRMs. For LRMs, majority voting outperforms complex methods (best-of-N, sequential revisions, etc) with minimal extra compute gains. It also reveals that LRMs’ correct responses are shorter, with fewer hedging/thinking markers but more discourse markers, which can improve ITC methods.

**Strengths:**

- The writing of the paper is clear and easy-to-follow.
- Personally, I appreciate this type of paper that conducts a unified comparison, analysis, and discussion of multiple TTS methods under rigorous experimental setups.

**Weaknesses:**

- The definition of trained-verifier-free TTS methods seems somewhat vague. The paper emphasizes that no external model is used as a verifier. However, in methods such as Sequential Revisions and Best of N, the model itself appears to be used as a verifier, with only a difference between internal and external verifiers.
- To my knowledge, the scoring ability of LLMs is far lower than their comparison ability. The Best of N method in the paper scores different responses separately, which may be less effective than comparing multiple responses simultaneously.
- What is the abscissa of Figure 4? Is it the same as that of Figure 3? It seems that there is no explanation in the figures.

**Questions:**

- This paper focuses solely on trained-verifier-free TTS methods. Why don’t you consider expanding to those with a verifier?
- Last but not least, I am somewhat curious about the performance of LRMs on more difficult datasets (e.g., HLE), whether majority voting might fail or be less effective than other methods.

---

### Note · Authors · 2025-12-03

I have read and agree with the venue's withdrawal policy on behalf of myself and my co-authors.